# Virtual Fitting Room: Generating Arbitrarily Long Videos of Virtual Try-On from a Single Image

**Jun-Kun Chen**[1,2*]   **Aayush Bansal**[1]   **Minh Phuoc Vo**[1]   **Yu-Xiong Wang**[2]

[1]SpreeAI    [2]University of Illinois Urbana-Champaign

immortalco.github.io/VirtualFittingRoom

## Abstract

We introduce Virtual Fitting Room (VFR), a novel video generative model that produces arbitrarily long virtual try-on videos. Our VFR models long video generation tasks as an auto-regressive, segment-by-segment generation process, eliminating the need for resource-intensive generation and lengthy video data, while providing the flexibility to generate videos of arbitrary length. The key challenges of this task are twofold: ensuring local smoothness between adjacent segments and maintaining global temporal consistency across different segments. To address these challenges, we propose our VFR framework, which ensures smoothness through a prefix video condition and enforces consistency with the anchor video—a 360° video that comprehensively captures the human's whole-body appearance. Our VFR generates minute-scale virtual try-on videos with both local smoothness and global temporal consistency under various motions, making it a pioneering work in long virtual try-on video generation.

## 1   Introduction

Imagine being in a fitting room, trying on a garment, when a hurried knock interrupts you. Would that allow you to truly experience the garment before buying it? No. To truly understand a garment, one may want to interact with it in various ways. The computational methods for virtually trying on a garment enable a user to see themselves, but only in an image [1–6] or a short 5∼10s video [7, 8], limiting the user's ability to fully experience a garment. We introduce *Virtual Fitting Room* (VFR) to enable a user to study the interaction of garments with their body as long as they like. Unlike existing image or video try-on methods, VFR allows a user to create arbitrarily long videos ($720 \times 1152$ resolution at 8 FPS and can be further refined to 24 FPS) of themselves, given a single user image, a desired garment, and a reference video performing the desired try-on motion. Fig. 1 shows a 30s and a 90s video generated using our method.

Generating a 5s-long video is already a computationally demanding task [7, 8]. Naively extending these methods requires even more computational resources and a large-scale video dataset containing long videos for learning. To overcome these limitations, one may generate multiple short segments of a long video one by one "auto-regressively" in timestamp order, and then merge them to create a long video, as visualized in Fig. 4-(a). Inspired by common approaches in general long video generation [9–14], we allow each segment to slightly overlap with the previous segment, and pass the overlapped "prefix" of the current segment as a condition, ensuring a locally smooth transition between each adjacent segment pair. However, these generated videos lack global temporal consistency, as shown in Fig. 2-(a), which is difficult to fix after once they are generated. In this work, we draw inspiration from the process of writing an essay with an outline as an "anchor." We posit that creating long videos is a two-step process that involves: (1) creating an "outline" or an "anchor" to guide the

---

*Work done during an internship at SpreeAI.

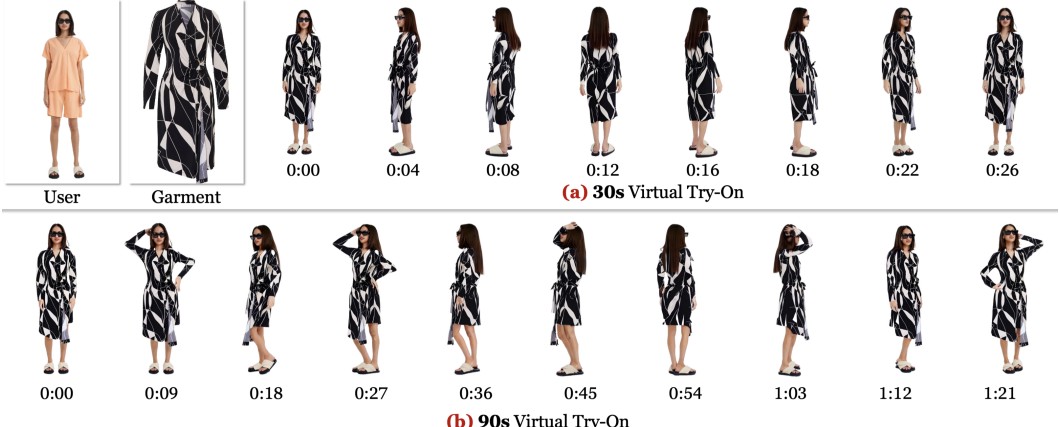

| User | Garment | 0:00 | 0:04 | 0:08 | 0:12 | 0:16 | 0:18 | 0:22 | 0:26 |

**(a) 30s** Virtual Try-On

| 0:00 | 0:09 | 0:18 | 0:27 | 0:36 | 0:45 | 0:54 | 1:03 | 1:12 | 1:21 |

**(b) 90s** Virtual Try-On

Figure 1: We generate two arbitrarily long videos: **(a)** a 30s video; and **(b)** a 90s video, for a user interacting with a given garment. Our approach preserves accessories – glasses and slippers, and allows a desirable user-garment interaction. Please refer to the project page for full streaming.

generation; and (2) generating multiple short videos that are consistent with the anchor. We observe that a short 360° video like Fig. 3-(a) of the human subject in a simple "A" pose serves as a reasonable anchor, allowing the model to comprehensively design the whole-body appearance of the human. The multiple short video segments are consistent with the anchor video and, therefore, are also consistent with each other. With the anchor video, our generated long video achieves temporal consistency (Fig. 2-(b)) *without* requiring long videos for training.

**Evaluating the quality of long video virtual try-on.** With the flexibility to generate arbitrarily long videos, we introduce an evaluation protocol to assess performance across four aspects with varying difficulty levels: (1) **360° Garment Consistency** – a 360° video of a stationary human subject in "A" pose, which allows us to study the quality of the generated garment (Fig. 3-(a)); (2) **30s Human+Garment Consistency** – a 30s video of a human subject casually moving around a point in front of a stationary camera, which enables us to assess the quality of the generated human and garment (Fig. 1-(a)); (3) **Hand-Body Interaction Faithfulness** – a 90s video of a human subject performing a fixed set of poses in front of a stationary camera, which facilitates the evaluation of the robustness of the virtual try-on method in controlled settings (Fig. 1-(b)); and (4) **Capability for Arbitrary Poses** - a 30~60s video of a human subject freely interacting with their body, which allows us to investigate robustness in various poses and orientations. We believe that this evaluation protocol will enable us to comprehensively assess the quality of virtual try-on methods.

*Free* **viewpoint rendering is for** *free*. A by-product of learning temporally consistent video is that we can render a human subject in any pose and viewpoint. Fig. 3-(a) shows a generated 360° anchor video ("A" pose) of a user wearing the target garment. The output is 3D consistent, enabling us to reconstruct it into a 3D mesh, as visualized in Fig. 3-(b) in a NeRFStudio [15] viewer. We observe that 3D implicitly emerges while enforcing temporal consistency. Interestingly, we can *recloth* and *reanimate* any human subject from a single image, and view them from any viewpoint.

**Our Contributions.** (1) We introduce VFR, the first method to generate arbitrarily long, high-resolution ($720 \times 1152$ resolution at 24FPS) human videos of virtual try-on from a single image. (2) We also introduce an evaluation protocol to assess the overall quality of virtual try-on methods. (3) We observe that the proposed method implicitly learns 3D consistency, enabling us to perform free-viewpoint rendering. Though designed for long video virtual try-on, our approach also provides insights for general long video generation, where the anchor mechanism offers a simple way to maintain global consistency over time.

## 2 Related Work

We believe *static 2D imagery isn't enough for a realistic virtual try-on experience*. The challenge of achieving high-resolution virtual try-on escalates as we progress from images to videos, and ultimately to 4D. This progression not only heightens the demand for computational resources, but

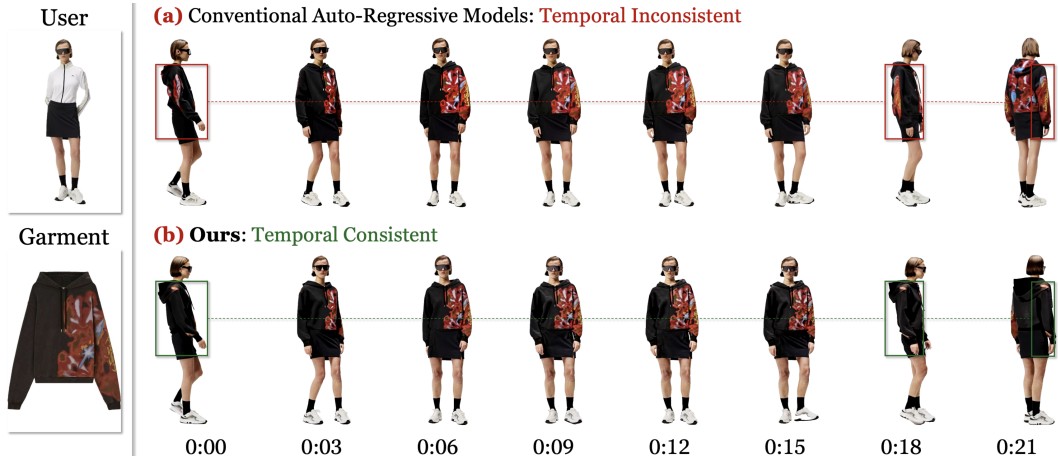

Figure 2: Given a user and target garment, **(a)** conventional auto-regressive video generators suffer from temporal inconsistency issues between distant frames. Note different patterns of sleeves in red bounding boxes across the time. **(b)** Our VFR generates temporally consistent try-on videos.

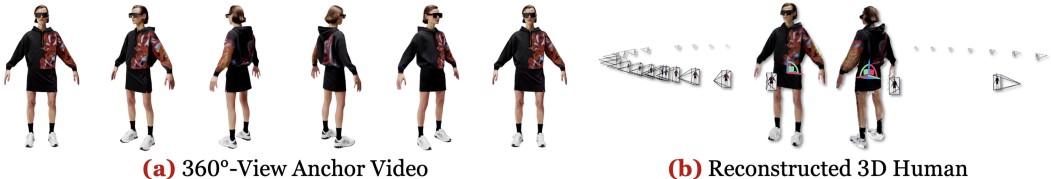

**(a)** 360°-View Anchor Video          **(b)** Reconstructed 3D Human

Figure 3: **(a)** VFR produces a 360° anchor video ("A" pose) of a user for a given garment. We observe that the outputs are 3D-consistent, allowing us to **(b)** reconstruct it into a 3D human mesh.

also diminishes the availability of previous extensive databases necessary for learning. Our goal is to identify computational methods that enable the generation of arbitrarily long videos, even with constrained resources.

**Image-to-Image Try-on.** Given an image of the user and a reference to the target garment, the goal here is to synthesize a new image of the user wearing the target garment [16–37, 4, 38–40]. Most methods take a two-step approach: (1) deform the garment for a given user, known as warping; and (2) generate a new image with the deformed garment using GAN [41] or latent diffusion models [42–44]. A few exceptions [20, 33, 45] generate the output without an intermediate warping step. The primary limitation in this field is the restricted experience a user can have with garments. One can only see them in exactly the same pose as in the input image. This issue can potentially be addressed by incorporating an additional module that can synthesize humans in different poses [46–51]. However, due to error propagation, the garment's appearance becomes inconsistent. Consequently, various methods [52–56] aim to jointly change the pose and the garment. These methods, however, lack temporal consistency and smoothness when applied across a series of poses.

**Video-to-Video Try-on.** Given a video of the user and a reference to the target garment, the objective here is to synthesize a new video of the user wearing the target garment [57–62]. An important distinction is ensuring that the synthesized garments and humans are temporally consistent and accurate. He et al. [63] employs an image-to-image try-on methodology, but incorporates an additional temporal loss during training to enforce consistency. Recent methods [7, 8] can generate high-resolution output, but they are limited by the duration of the generated video ($5s$). Naively increasing the duration of videos will require enormous computational resources.

**Image-to-Video Try-on.** In this work, we explore the generation of arbitrarily long videos from a single user image and garment images. A naive approach is to utilize a single image try-on method and animate it using an image-to-video creation module [64–76]. However, a modular approach leads to error propagation, which degrades the quality of the generated outputs. Therefore, we seek an end-to-end video generation pipeline that allows us to preserve the details of the garment [8, 77]. We observe that text conditioning cannot effectively capture long and subtle movements [8]. Instead, we use example videos as a reference to guide the creation of new videos. A notable prior work,

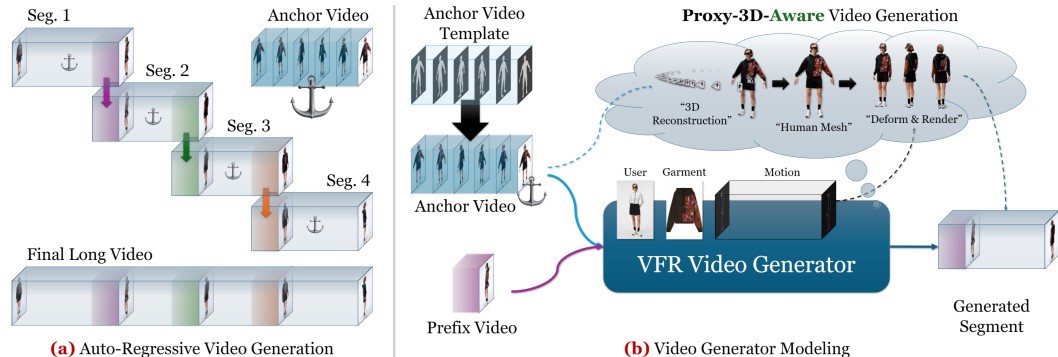

Figure 4: **(a)** Our VFR is an auto-regressive framework that generates a long video segment-by-segment. **(b)** The video generator takes both an anchor video and a prefix video as input, and generates a new segment that continues the prefix video while maintaining consistency with the anchor video.

Dress&Dance (DnD) [8], generates 5s videos with high resolution $720 \times 1152$ at 24FPS. In this work, we generate arbitrarily long videos from a single user image and the reference garments.

**Long Video Generation.** There is also a line of work [9–14] that investigates long video generation for text-to-video and image-to-video tasks. The common idea behind these methods is to introduce an additional "memory back" or "history tracking" mechanism to ensure consistency with the previous frames in a typical auto-regressive generation process. For example, a concurrent work, FramePack [14] designs a computationally efficient way to consider all the previous frames as conditions when generating a new frame. Similarly, StarGen [78] proposes a spatio-temporal autoregression framework for static scene synthesis, while LCT [79] introduces a long-context tuning strategy that leverages interleaved 3D positional embeddings and KV-cache mechanisms to extend temporal receptive fields. These approaches can generate smooth long videos; however, their temporal consistency is not guaranteed and often violated due to ineffective history tracking or over-compressed memory. In contrast, our VFR method tackles this problem by designing an anchor video conditioning mechanism tailored for virtual try-on tasks, providing a global appearance prior that promotes strong temporal consistency across the entire long video.

**Free Viewpoint Rendering.** Our ability to generate arbitrarily long videos implicitly allows us to perform free-viewpoint rendering of humans [80, 81]. We observe that a model trained to capture consistent temporal characteristics implicitly learns 3D consistency in its outputs. This analysis could potentially pave the way for 4D try-ons in the future.

## 3   VFR: Methodology

Given a user image, a reference garment image, an optional text prompt, and a long motion reference video, our VFR is a method that generates long, minute-scale, high-quality try-on videos of the user wearing the desired garment and performing the indicated motion, in an auto-regressive, segment-by-segment manner. In an auto-regressive generation framework, the core challenge is to achieve both (1) *local smoothness* such that the video transitions seamlessly without noticeable sudden changes or morphing (Fig. 5); and (2) *global temporal consistency* such that the appearance of both the user and the garments at all occurrences in the video is the same (Fig. 2).

To address these crucial challenges, our key insight is to (1) propose the "anchor video" generation to ensure a consistent appearance throughout the entire video, and (2) introduce the video prefix condition and immediate refiner to enhance video smoothness through strong conditioning. With both insights, our VFR achieves high-quality long try-on videos, while ensuring both smoothness and temporal consistency.

**Auto-Regressive Video Generation.** There are two approaches to generating long videos: one can either produce the entire long video in a single forward pass, just like generating a short video, or generate the video auto-regressively as multiple shorter segments (Fig. 4-(a)). The former method necessitates significant computational resources during the generation process, as well as extensive long video datasets; whereas the latter requires similar computational resources and training datasets to those used for short video generation.

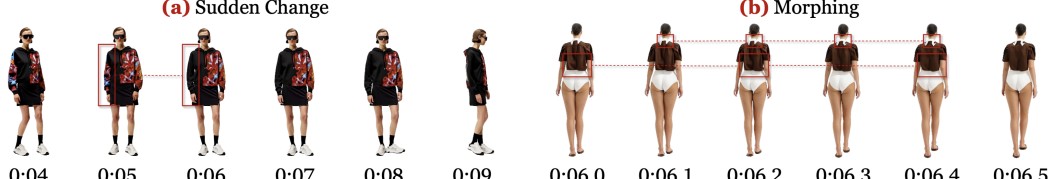

(a) Sudden Change      (b) Morphing

0:04   0:05   0:06   0:07   0:08   0:09     0:06.0   0:06.1   0:06.2   0:06.3   0:06.4   0:06.5

Figure 5: Without our prefix conditioning, the generated video may contain artifacts like **(a)** sudden changes or **(b)** morphing, as highlighted in the boxes, which violate the smoothness requirements.

We opt for the second approach and design VFR as an *auto-regressive* generative model $v \leftarrow G(c, v_{\text{prefix}})$. As seen in Fig. 4-(b), our model generates a short $F$-frame video $v$ with input conditions $c$, such that the generated video has $v_{\text{prefix}}$ as its prefix, and then seamlessly continues into the preceding frames. In long video generation, we first generate the initial $F$ frames $v_0 \leftarrow G(c_0, \varnothing)$, and then generate each $(F - f)$-frame segment $v_i \leftarrow G\left(c_i, v_{i-1}^{[F-f+1,F]}\right)$, using the $f$ overlapped frames $v_{i-1}^{[F-f+1,F]}$ from the previous segment as its prefix. The reason for using $f$ frames instead of just one frame is to ensure that both the visual appearance and the motion, which requires multiple frames to represent, are smoothly continued into the next segment $v_i$. To obtain the different conditions $c_i$ for each segment $v_i$, we only need to replace the full motion reference video with its corresponding segment while keeping all other conditions the same. The final generated video is $v = v_0 + \text{Concat}_{i=1}^{n} v_i^{[f+1,F]}$ (Fig. 4-(a)). Please refer to the Supplementary for the model details.

## 3.1 Anchor Video for Temporal Consistency

The issue of temporal consistency arises from the ambiguity of the unseen aspects of the garment or the user's appearance. For instance, given only a frontal image of the garment, its back and side views can be interpreted in multiple, yet plausible, ways. This leads to temporal inconsistencies when different appearances are generated in two non-overlapping segments, as shown in Fig. 2-(a). To ensure consistency, a naive approach is to use the prefix of the whole video $v_i^{\text{pre}} = v_0 + \text{Concat}_{j=1}^{i-1} v_j^{[f+1,F]}$ as part of the condition $c_i$ to generate the next segment $v_i$, such that the generated $v_i$ remains consistent with any frame that has appeared in $v_i^{\text{pre}}$. However, this approach negates all the advantages of the auto-regressive generative framework – it also incurs significant computational costs and requires long videos for training to effectively handle $v_i^{\text{pre}}$.

Our insight is to organize the appearance information of garment and user in a more effective representation than the whole $v_i^{\text{pre}}$. Therefore, we introduce the concept of the anchor video $v_a$, which is a short $F_a$-frame video of the user wearing the garment while performing a different motion, ensuring that any part of the user and the garment is visible in at least one of its frames. Given a motion template $m_t$, the generation of the anchor video can be modeled as $v_a \leftarrow G(c|_{\text{motion}=m_t}, \varnothing)$, i.e., generating a virtual try-on video with fixed motion $m_t$.

We choose $m_t$ to be a stationary A-pose like Fig. 3-(a), with both arms slightly raised near the sides of the waist and legs naturally apart, under a 360°-view camera trajectory. As shown in Fig. 3-(b), such an anchor video generated from this $m_t$ is already sufficient for reconstructing a 3D mesh representation with NeRFStudio's NeRFacto model [15], indicating that this video effectively captures a 3D-consistent appearance of the try-on human video. Conditioning the generator on this video anchor is sufficient to promote consistency across poses and viewpoints, ie., a proxy-3D-aware video generation, without any explicit 3D representation [82] or 3D motion modeling like SMPL [83]. Since consistency is an equivalence relationship, each $v_i$'s consistency with $v_a$ implies their consistency with one another. As a result, we achieve cross-segment global appearance consistency, which guarantees the temporal consistency of the final generated long video, as shown in Fig. 2-(b).

## 3.2 Auto-Regressive Prefix Conditioning for Smoothness

In our auto-regressive generation, we aim to generate the next segment $v_i$, such that its first $f$ frames $v_i^{[1,f]}$ precisely match the given $v_{\text{prefix}}$. If we generate an arbitrary video $v_i$ and then replace its first $f$ frames, there may be a sudden change between the $f$-th and $(f + 1)$-th frame, resulting in a discontinuous video. A straightforward approach is to model this task as "outpainting" the remaining

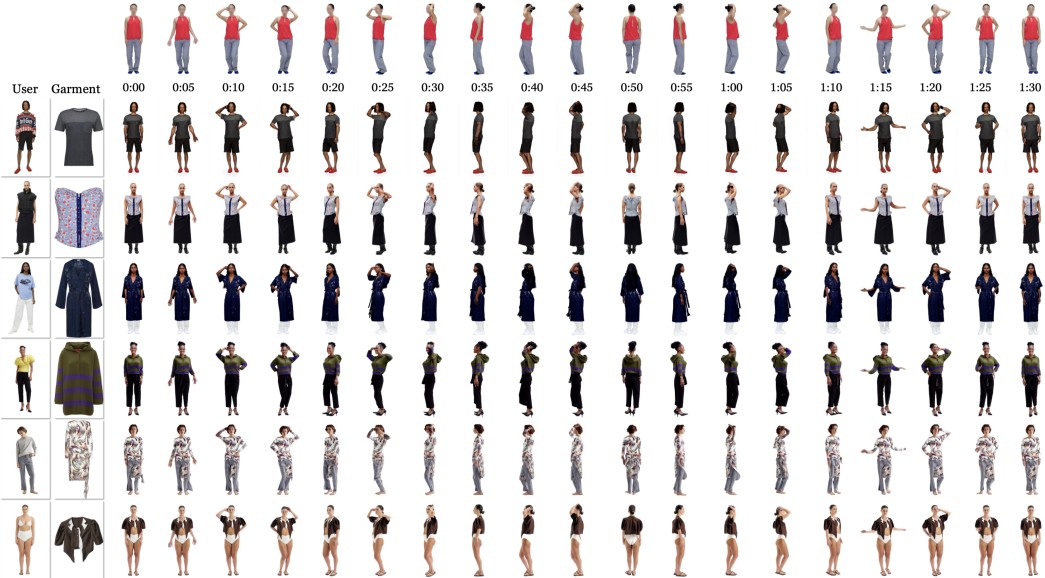

Figure 6: In the virtual try-on with the 90s hand-body interaction motion, our VFR generates temporally consistent try-on videos. Please refer to the project page for full streaming.

$(F - f)$ frames, given the first $f$ frames. This allows us to apply a training-free inpainting/outpainting method, DiffEdit [84], leveraging the same video diffusion model to generate a visually smooth video that starts from the given prefix. However, such a generated video may still contain sudden changes (Fig. 5-(a)) and morphing between different appearances (Fig. 5-(b)).

To enforce the video diffusion model to generate smooth videos, we must explicitly consider the fixed prefix $v_{\text{prefix}}$ during the generation process. Therefore, as demonstrated in Fig. 4-(b), we introduce the prefix as a conditioning input. This approach ensures that the model can learn to generate a smooth video segment that seamlessly continues the input prefix.

### 3.3 Training Long Video Generator with Short Video Data

The introduction of auto-regressive generation focuses on each $F$ frame segment of the video, which only requires videos containing at least $F$ frames for training. This approach eliminates the need for large-scale, lengthy video training data, which is challenging to obtain, especially for our virtual try-on videos. Each single generation can be modeled as $v \leftarrow G(c, v_{\text{prefix}})$, where $c = (c_{\text{try-on}}, v_a)$ represents both try-on conditions (garment and user image, motion reference video, etc.) and the anchor video. For each $F$-frame segment $v$ in our dataset, we can set $v_{\text{prefix}} \leftarrow v^{[1,f]}$ to obtain the corresponding input. However, in most cases, we do not have the desired anchor video $v_a$ that is exactly at the $m_t$ we want. Generating $m_t$ by ourselves may also be inconsistent with the video, e.g., with different back or side views.

To address this issue, our approach does not focus on the $v_a$ at a specific $m_t$, but instead allows the model to learn to leverage $v_a$'s under an *arbitrary* motion $m_t$. This ensures that, regardless of the pose of the $v_a$, the video generator can always optimize its usage to produce videos that are consistent with the given $v_a$. With this modeling, *any* video segment showing the same person wearing the same garment can be used as an anchor video for the target video $v$. More specifically, for each $F_{\text{raw}}$-frame raw video data, we can sample two intervals – with the same user and garments – to construct a training batch: a $F$-frame interval as target $v$, and another *arbitrary length* interval to temporally upsample or downsample it to $F_a$ frames and use as $v_a$. Utilizing this training strategy, we can effectively train the model with only short ($F \sim 3F$) raw videos, which is not significantly longer than $F$, yet can already yield a sufficient variety of different pairs of $v$ and $v_a$.

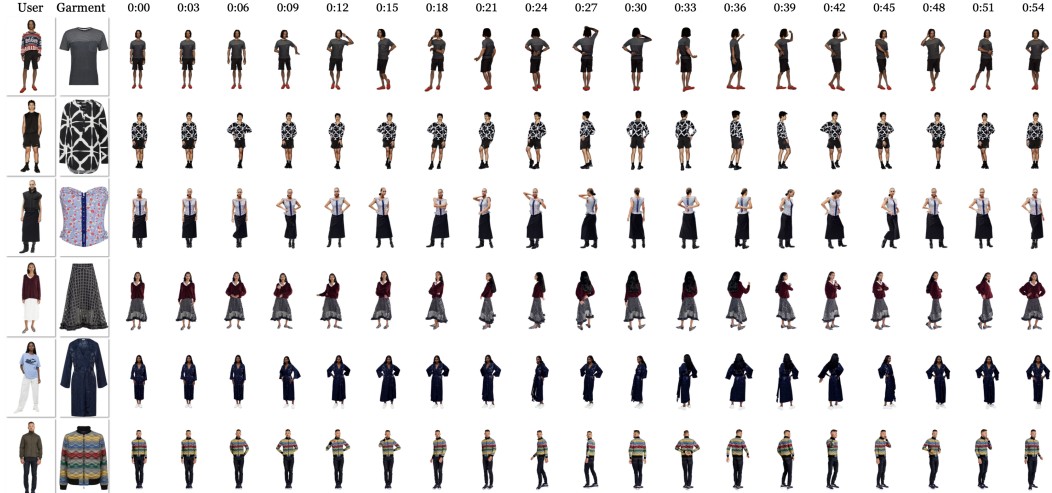

Figure 7: In the virtual try-on with a ∼50s arbitrary motion, our VFR faithfully preserves consistent garment details and human appearance, showcasing various poses with high quality. These results are shown as videos on our project page.

## 3.4 Immediate Refinement of the Generated Segments

The robustness of auto-regressive video generation is significantly more critical than that of end-to-end generation, as artifacts and errors can accumulate and amplify over time. Specifically, given that both anchor and prefix video conditions are complex, pixel-unaligned video conditions, we observe increasing artifacts in our initial generation results, which may accumulate to severe degradation.

Inspired by Dress&Dance [8], which leverages a standalone generator as a "refiner" to temporally upsample the video and remove artifacts, we also design a refiner for our generation. In our case, we do not aim to upsample the videos; instead, we focus entirely on artifact removal and video smoothing. Our refiner can be modeled as $v' \leftarrow G_{\mathrm{refiner}}(c, v_{\mathrm{prefix}}, v)$, which takes a $F$-frame video $v$ as input, along with all the other conditions $c$, including $v_a$, obtaining a refined $F$-frame video $v'$. Such a refiner is directly applied to each $v_i$, immediately after its initial generation, so we call it an "immediate refiner." Additionally, since the input $v$ and output $v'$ have the same number of frames, we apply the refiner $G_{\mathrm{refiner}}$ multiple times to the same video to ensure complete artifact removal. The final output segment $v_i$ can be expressed as $v_i \leftarrow G_{\mathrm{refiner}}^{\circ k}\left(c, v_{i-1}^{[F-f+1,F]}, G\left(c, v_{i-1}^{[F-f+1,F]}\right)\right)$, where $k$ is the number of iterations. With this immediate refinement technique, we resolve the issue of degradation over time in long video generation, maintaining high quality from beginning to end.

# 4 Experiments

## 4.1 Experimental Settings

**Model Training Settings.** Our VFR model is built on Dress&Dance [8] with the addition of "prefix video" and "anchor video" CondNets. We train VFR on both Internet and captured datasets from Dress&Dance for $10,000$ iterations. Specifically, the immediate refiner is initialized from the $5,000$-th iteration checkpoint of the base VFR, and trained for another $5,000$ iterations.

**Evaluation Tasks.** We evaluate VFR on four different parts: (1) **360° Garment Consistency**, to generate a 5s 360°-view "A" pose video, which also serves as the anchor videos for the other tasks; (2) **360° Human+Garment Consistency**, to generate a 30s 360° casually moving video; (3) **360° Hand-Body Interaction Faithfulness**, to generate a 90s video with a fixed motion; and (4) **Capability for Arbitrary Poses**, to generate a 30∼60s video with arbitrary motion. These benchmarks are available on our project page. Notably, existing public video try-on datasets VVT [85] and ViViD [61] are not suitable for evaluating our setting, as their spatial resolutions (256×192 and 832×624, respectively) and temporal lengths (typically only a few seconds) are insufficient to capture the high-resolution, minute-scale, and temporally consistent generation capabilities targeted by our benchmark.

**Baselines.** We compare our VFR against the baselines **FramePack** [14], utilizing an "image virtual try-on + image-to-video animation" (IVT+I2V) procedure that is aligned with Dress&Dance [8]. We also compare with **Kling Video 2.0** [86] in a "repeating image-to-video" (RI2V) manner mentioned in [14]. We are unable to compare with the following baselines: StreamT2V [13], as it only supports 16:9 landscape videos; CausVid [12], given that there is no available image-to-video checkpoint released; TTT [9], since it only supports Tom and Jerry videos; and DiffusionForcing [10] and HistoryGuidance [11], as they are restricted to videos from their respective trained datasets. As for the image try-on method, we mainly use the two state-of-the-art method, **Dress&Dance** image try-on [8] and **Kling Try-On** [86], to generate the first frame of the video. More combinations including other image virtual try-on methods are compared in our Supplementary.

**Ablation studies.** We also compare our full VFR with the following variants: (1) **"No Prefix" (NP)**, which does not use prefix conditioning, but directly utilizes DiffEdit [84] to outpaint the video with the prefix; (2) **"No Anchor" (NA)**, which does not generate and condition the segment generations on the anchor video; (3) **"No Refine" (NR)**, which skips the immediate refinements after each segment generation; (4) **"Dress&Dance" (D&D)**, which does not use either the anchor video or the prefix conditioning, making it equivalent to a training-free method that employs Dress&Dance with DiffEdit for long video generation.

**Metrics.** Consistent with Dress&Dance [8] and FramePack [14], we utilize GPT [87]-based scores and VBench [88–90] to evaluate our videos. GPT scores can effectively assess the try-on quality from various aspects, leveraging GPT's visual capabilities; VBench introduces a set of metrics that comprehensively evaluate the videos from both quality and semantic perspectives.

## 4.2 Experimental Results and Analysis

We present our qualitative results in Figs. 6,7,8 as images, and on our project page as videos.

**360° Human+Garment Consistency (30s) – Fig. 8.** As shown Fig. 8-(a), our VFR produces high-quality, long virtual try-on videos. In Fig. 8-(b), our "No Prefix" (NP) variant, due to the global anchor videos, generates results that are comparable to our full VFR, but exhibits some sudden changes, as illustrated on our project page. In contrast, our "No Anchor" (NA) variant's video displays long-term temporal inconsistencies, while our "Dress&Dance training-free" (D&D) variant exhibits even greater temporal inconsistencies in both the short and long term. This highlights that both designs in our VFR for local smoothness and global temporal consistency are effective and essential. In Fig. 8-(c), the baseline FramePack [14] produces overall smooth results with long-term inconsistencies, while the appearances deviate significantly in the results produced by the Kling Video 2.0 [86]-based RI2V method. This demonstrates that the virtual try-on tasks are non-trivial and challenging, underscoring our contribution in achieving high-quality results.

**Hand-Body Interaction Faithfullness (90s) – Fig. 6.** The motion in these evaluation tasks encompasses both human rotation and arm movements. Our VFR faithfully performs the same motion in different virtual try-on tasks, demonstrating the ability to depict the same motion across various garments – either pants, skirts, or dresses. Even for such a long-term video, our VFR still maintains high temporal consistency, which can be observed by comparing the first and the last frame.

**Capability for Arbitrary Poses (~50s) – Fig. 7.** In these highly challenging tasks, the motions can be arbitrary, encompassing various arm and leg movements, which lead to even more diverse showcases and interactions between garments and users. We observe that our VFR effectively handles these motions and generates high-quality visualizations to depict them. This shows VFR has the capability to generalize to various long video virtual try-on tasks.

**Quantitative Analysis.** The quantitative evaluation comparisons are provided in Table 1. We use the "Subject Consistency," "Background Consistency," and "Motion Smoothness" from VBench [88] to evaluate how the two major challenges – temporal consistency and smoothness – are addressed, as well as the GPT metric in [8] to assess the overall virtual try-on quality. We provide the more quantitative evaluations in Supplementary.

As shown in Table 1, in each component of the evaluation protocols, our full VFR consistently outperforms all the baselines and variants. Furthermore, since our VFR is based on the previous work Dress&Dance [8], all these variants maintain a portion of its virtual try-on capability, achieving

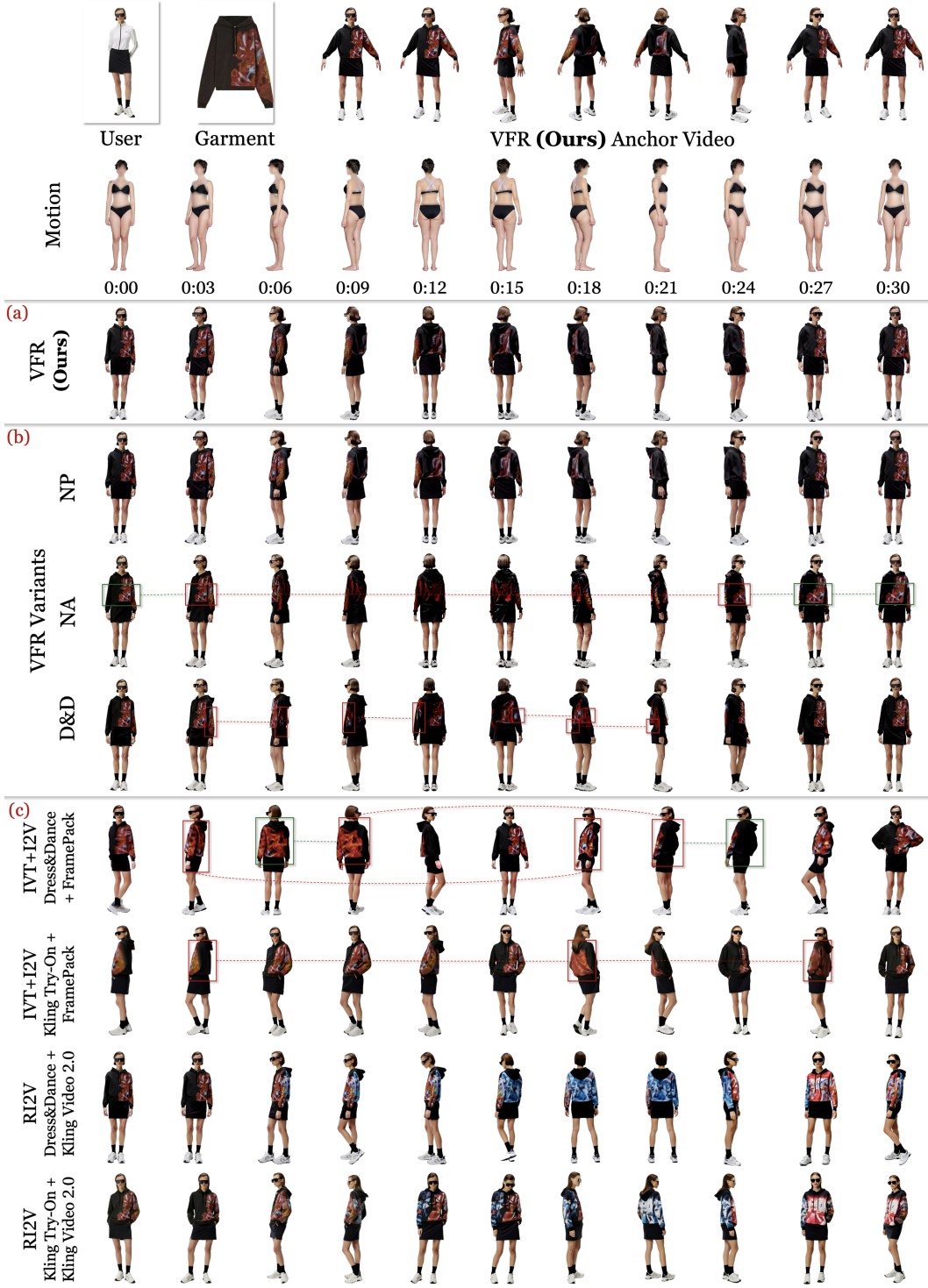

Figure 8: In the virtual try-on with 360° dynamic motion, (a) our VFR generates high quality, long virtual try-on videos, while (b) removing either anchor video or prefix conditioning results in noticeable degradations. On the contrary, (c) the baselines suffer from smoothness and temporal consistency issues. Please refer to the project page for full streaming.

comparable GPT evaluation scores, where the D&D variant, as a training-free long video generation pipeline utilizing D&D, preserves the majority of its capability.

We also observe that our NP variant performs similar to our full model, reflecting the strong control provided by the anchor video. However, without the anchor video the NA and D&D variants suffer a

| Method | | Subject Consistency | Background Consistency | Motion Smoothness | GPT$_{\text{Try-On}}\uparrow$ | GPT$_{\text{User}}\uparrow$ | GPT$_{\text{Motion}}\uparrow$ | GPT$_{\text{Visual}}\uparrow$ | GPT$_{\text{Overall}}\uparrow$ |
|---|---|---|---|---|---|---|---|---|---|
| **1. 360° Garment Consistency (5s)** | | | | | | | | | |
| Ours | | 92.84 | 95.38 | 98.35 | 87.83 | 85.90 | 76.67 | 80.35 | 82.06 |
| **2. 360° Human+Garment Consistency (30s)** | | | | | | | | | |
| Ours | | **94.06** | **96.53** | **99.37** | **90.09** | **88.11** | 84.08 | **86.33** | **87.14** |
| Ours | NP | 93.58 | 96.10 | 99.31 | 89.66 | 87.24 | **84.20** | 85.86 | 86.69 |
| Ours | NA | 92.77 | 95.55 | 99.22 | 88.47 | 84.91 | 81.75 | 81.66 | 84.04 |
| Ours | D&D | 93.70 | 96.01 | 99.23 | 89.72 | 86.16 | 83.20 | 86.05 | 86.10 |
| Ours | NR | 90.80 | 94.82 | 99.21 | 87.29 | 85.55 | 78.74 | 71.90 | 80.40 |
| **3. Hand-Body Interaction Faithfulness (90s)** | | | | | | | | | |
| Ours | | **92.13** | **94.02** | **99.24** | 87.20 | 84.88 | 77.85 | 81.12 | 82.62 |
| Ours | NP | 91.88 | 93.91 | 99.15 | 89.65 | 86.28 | 80.15 | 83.22 | 84.62 |
| Ours | NA | 88.00 | 91.50 | 99.09 | 85.20 | 80.35 | 70.67 | 69.05 | 75.70 |
| Ours | D&D | 91.65 | 93.25 | 99.11 | **89.95** | **87.25** | **82.53** | **85.15** | **86.15** |
| Ours | NR | 86.02 | 89.84 | 99.01 | 85.22 | 84.83 | 72.25 | 65.10 | 75.03 |
| **4-Med. Capability for Arbitrary Poses – Medium (25s)** | | | | | | | | | |
| Ours | | **91.09** | **93.82** | **98.62** | 87.11 | 84.45 | 77.83 | 79.77 | 81.93 |
| Ours | NP | 90.19 | 93.00 | 97.85 | 87.40 | **85.48** | 78.40 | 80.55 | 82.56 |
| Ours | NA | 88.46 | 92.10 | 97.92 | 86.87 | 84.39 | 77.92 | 77.56 | 81.31 |
| Ours | D&D | 89.27 | 92.81 | 97.83 | **87.83** | 85.23 | **80.24** | **82.22** | **83.61** |
| Ours | NR | 87.56 | 91.79 | 97.76 | 85.28 | 83.75 | 74.85 | 68.98 | 77.26 |
| **4-Hard. Capability for Arbitrary Poses – Hard (30-50s)** | | | | | | | | | |
| Ours | | **93.21** | **95.29** | **99.35** | 87.03 | 85.83 | 69.99 | 78.84 | 79.05 |
| Ours | NP | 92.71 | 94.63 | 99.29 | 87.69 | **86.38** | 71.12 | 79.47 | 79.60 |
| Ours | NA | 91.17 | 93.31 | 99.22 | 87.14 | 85.03 | 71.42 | 77.11 | 78.99 |
| Ours | D&D | 92.27 | 94.09 | 99.25 | **88.05** | 84.97 | 71.90 | **81.38** | 80.39 |
| Ours | NR | 88.17 | 92.08 | 99.19 | 85.36 | 84.61 | 67.45 | 64.55 | 73.44 |
| IVT+I2V | D&D + FramePack [14] | 88.33 | 90.98 | 98.24 | 86.96 | 85.35 | **77.51** | 79.33 | **81.86** |
| RI2V | D&D + Kling [86] | 92.71 | 94.08 | 98.92 | 87.25 | 85.59 | 65.80 | 79.40 | 77.24 |

Table 1: Quantitative experiments show that our VFR consistently outperforms all variants and baselines in both consistency and smoothness metrics from VBench [88], while achieving comparable try-on and visual quality as Dress&Dance (D&D) [8]. `Notations:` NP is No Prefix; NA is No Anchor; and NR is No Refine.

significant drop in both (human) subject and background consistencies, and the degradation of virtual try-on quality even occurs for the NA variant, as reflected in the GPT evaluation metric. This shows that the anchor video controls not only the consistent appearance but also the try-on quality.

Finally, we compare the most powerful baselines, FramePack [14] and Kling [86], combined with the state-of-the-art virtual try-on method Dress&Dance. FramePack has a significantly lower consistency metric, indicating that its conditioning method, which takes into account the previous frames, is not sufficient to enforce consistency. Kling still achieves a slightly lower consistency metric compared to our VFR, but the quality of the virtual try-on degrades.

# 5   Conclusion

We propose VFR, a virtual try-on method that generates arbitrarily long, high-resolution videos from a single user image, garment, and motion reference. The key of our method is an anchor video-guided framework that ensures temporal consistency across segments and implicitly captures 3D structure, enabling free-viewpoint rendering without 3D supervision. Along with the prefix conditioning, we achieve both local smoothness and global temporal consistency in the long video generation. We further introduce a new evaluation protocol tailored to long video virtual try-on, covering garment fidelity, user appearance, and motion robustness. Experiments show that VFR produces realistic, smooth, temporally consistent, and garment-faithful results that significantly surpass the capabilities of prior methods. We believe that VFR opens up new avenues for interactive, personalized virtual try-on experiences—whether in e-commerce, virtual social platforms, or creative content generation.

**Discussions.** We made progress in generating arbitrarily long, high-resolution videos for virtual try-on. Our preliminary analysis shows that additional video data can help us further improve the quality of the generated videos. Secondly, while this work is the first of its kind, it takes 1~2 hours to generate a 30s video, which is not efficient enough. We leave this speed-up as an interesting future work. Finally, we believe that our work will pave the way for the transition from long videos to arbitrary 4D content, where a user can both change camera perspective and motion.

**Potential Societal Impacts.** The positive societal impacts of our VFR may include (1) revolutionizing the online shopping experience for clothing, (2) decreasing returns and replacements of clothes through improved pre-sale understanding, and (3) leading to an increase in both the number and volume of online clothing shops. On the other hand, VFR is inherently a model that produces human videos, and also brings the risks to produce biased, unethical, or unsafe results.

**Acknowledgement:** We thank Pixophilia Technologies Private Limited, India, and VietTechTools Co., LTD, Vietnam, for generously providing the visual data for our benchmark.

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

# Supplementary Material

## A    Supplementary Video (SV)

We provide a supplementary video (SV) as this YouTube video to visualize our videos and comparisons against baselines. Here is a table of contents of the SV:

- SV0:17 360° Garment Consistency (Anchor Videos).
- SV0:28 360° Human+Garment Consistency (30s).
- SV1:27 Hand-Body Interaction Faithfulness (90s).
- SV2:57 Capability for Arbitrary Poses – Medium (25s).
- SV3:23 Capability for Arbitrary Poses – Hard (30-50s).
- SV4:56 Comparison with VFR Variants (Ablation Study).
- SV6:49 Comparison with Baselines (speeded-up to the same duration).
- SV7:23 3D Reconstruction.

## B    Implementation Details

### B.1    Base Model and Architecture

Our VFR is built on Dress&Dance (D&D) [8], a prior work that generates high-resolution, 5-second virtual try-on videos.

D&D proposes CondNets for unified conditioning. Our VFR introduces two different conditions: an anchor video and a prefix video. These conditions are passed to the diffusion transformer (DiT) [93] model with two additional CondNets.

### B.2    Issues in CondNets

While the conditions in conventional D&D are either images or pixel-aligned videos (motion or refinement source), the conditions in VFR include the anchor video (pixel-*unaligned* videos) and the prefix video (pixel-aligned, but with a *shorter* duration). These conditions are significantly more challenging for conditioning – when generating each frame, the generator must consider the entire condition video instead of only the corresponding frame.

We observe that CondNets continue to perform well for the anchor video in an out-of-the-box manner. However, when the prefix video is present, it is common for the continued video to exhibit artifacts, such as floating dust, as demonstrated in the "No Refine" variant in SV4:56. This necessitates the use of the immediate refiner to remove such artifacts.

### B.3    Training Procedure

We train our VFR on $8\times$ NVIDIA H100 GPUs using a traditional PyTorch Distributed Data Parallel (DDP) framework. As mentioned in the "Experiments" section in the main paper, both the main generative model and the refiner are trained for 10,000 iterations on both Internet and captured datasets from D&D [8].

In order to learn how to remove the specific dusty artifacts caused by the prefix conditions in the refiner, we employ a new data augmentation method. We generate a number of ovals with random sizes, positions, orientations, time intervals, and velocities, and then manually create artifacts in these regions by either increasing or decreasing brightness, or replacing them with other regions to create residual artifacts. With this procedure, our immediate refiner is capable of effectively removing various and aggressive artifacts – which can be referred to as the artifacts in the "No Refine" variant's results in SV4:56.

### B.4 Generation Setting

The total number of frames $F$ generated by VFR is 41, at a rate of 8 frames per second, resulting in a 5s video. In the generation of non-first segments, we select $f = 9$, which corresponds to a 1s overlap between the current segment and the previous segment. Therefore, after generating the first segment of $F = 41$ frames (5s), we produce $F - f = 32$ new frames (4s) at each of the following segments. The cutting points of the video are at the $(4k + 5)$-th second, *i.e.*, 0:05, 0:09, 0:13, 0:17, *etc.* Without prefix condition, *i.e.*, the "No Prefix (NP)" variant, these positions are most likely to have sudden changes or morphings.

## C  Additional Experimental Results

### C.1  Additional Image Try-On Model for Baselines

We primarily use Dress&Dance [8] image try-on as part of the baseline model for our experiments. Additionally, we also employ Kling try-on [86] and CatVTON [40] to establish some other baselines. As shown in **SV**6:49, although these methods generate reasonable virtual try-on images, the limitation of the animation models (FramePack or Kling in "repeat I2V" mode) still prevents them from achieving temporal consistency.

### C.2  Ablation Study: Qualitative Experiments

The qualitative ablation studies are presented in **SV**4:56. From the video, we can clearly observe that the "No Prefix (NP)" variants exhibit highly noticeable sudden changes during the cut, highlighting the necessity of the prefix conditioning. Furthermore, for the "No Anchor (NA)" and "D&D" variants, the performance even degrades over time, emphasizing the importance of the anchor videos to stabilize the overall appearance and quality. Finally, without the immediate refiner, the NR variants show noticeable artifacts that accumulate over time. This demonstrates that the immediate refiner is an effective tool to prevent artifact accumulation and ensure robust long video generation. In practice, we adopt a fixed two-step lightweight refinement strategy, where each refinement round takes approximately 15–20 minutes on a single GPU, resulting in a total refinement time of about 30–40 minutes—roughly 25% of the overall generation time, with the main generation requiring around 80–90 minutes. Overall, the "No Refine (NR)'" variant exhibits mild background artifacts and boundary noise toward the end of long sequences while maintaining strong identity and garment consistency; the refinement effectively mitigates these degradations and enhances the temporal stability of the generated videos.

### C.3  Limitations of Metrics

As shown in **SV**4:56, there are indeed notable smoothness and consistency issues present in the variants and baselines, even for the most powerful variant NP. However, these metrics – which report very similar numbers – do not adequately reflect such discrepancies. Additionally, the GPT metric, which evaluates only the try-on quality, are also not effectively assessing our long virtual try-on generation quality, and they do not capture the consistency of the garments.

## D  3D Reconstruction

As mentioned in the main paper, the anchor videos generated by VFR are almost 3D consistent, and therefore can be reconstructed into 3D shapes. We perform some of this reconstructions. As shown in **SV**7:23, even if the videos are not perfectly 3D consistent, we can still obtain a noisy yet reasonable 3D shape. Therefore, with advanced 2D-to-3D distillation techniques like SDS [94], one might be able to actually obtain a precised a mesh and use it for rendering and deformation – which is implicitly achieved within our VFR.

Notably, being perfectly 3D consistent is *not* a requirement of VFR. In fact, in the "Methodology" section of the main paper, VFR is trained to be sufficiently robust to support an *arbitrary* video as the anchor video. It is also frequently the case that the generated anchor video has a stationary background while the person is rotating (**SV**0:17) – this kind of video also works well and is even

more desirable, as the background is an integral part of the anchor that needs to remain consistent throughout the video. The 3D reconstruction results in **SV**7:23 are constructed from the video after background removal.

