# OpenReview forum: "Virtual Fitting Room: Generating Arbitrarily Long Videos of Virtual Try-On from a Single Image"
_NeurIPS.cc/2025/Conference — NeurIPS 2025 poster_

### Official Review · Reviewer_6KVB · 2025-06-26

**Clarity:** 3
**Significance:** 2
**Originality:** 2
**Rating:** 4
**Confidence:** 4

**Summary:**

This paper introduces Virtual Fitting Room (VFR), a novel video generative model capable of synthesizing arbitrarily long virtual try-on videos. VFR generates extended videos autoregressively by producing segments sequentially, ensuring temporal coherence. To maintain smoothness and consistency, the model conditions each segment on both a prefix video (previously generated content) and an anchor video (reference appearance). Experimental results demonstrate that VFR surpasses existing baselines in performance.

**Questions:**

None.

**Ethical Concerns:**

["NO or VERY MINOR ethics concerns only"]

**Final Justification:**

My concerns are addressed during the rebuttal phase. Thus I am raising my score to positive.

**Limitations:**

yes.

**Quality:**

2

**Strengths And Weaknesses:**

Strengths:
1. The paper is well written and easy to follow.
2. The idea of using an anchor video to improve consistency is effective and interesting.

Weaknesses:
1. The technical novelty of the proposed method appears limited, as it primarily combines existing techniques.
2. I don't think it is appropriate to claim "implicit 3D-aware" generation, which also appears confusing in Figure 4. Any well-trained video generative model inherently exhibits such properties to some degree.

---

> ### Author Rebuttal · Authors · 2025-07-31
>
> > The technical novelty of the proposed method appears limited, as it primarily combines existing techniques.
>
> - We respectfully disagree with this assessment.
>   - Our VFR is, to our knowledge, the first framework designed specifically for **long video virtual try-on**. Prior work on video try-on (L83, L90; e.g., D\&D, ViViD, Fashion-VDM) focuses on **short clips**, while prior work on long video generation addresses **non-try-on** domains (L99), such as text/image-to-video generation (e.g., FramePack) or static scene rendering (e.g., StarGen, CVPR’25).
>   - Our core contributions include the **anchor video mechanism** and the **training scheme for temporal consistency over long durations with short video data**—both of which, to our knowledge, are novel in the context of both try-on and (long) video generation.
>   - Notably, prior long-video methods (e.g., FramePack, StarGen, LCT) emphasize local smoothness but do not address **global temporal consistency** (L104), which is explicitly tackled by our anchor video design, tailored for the try-on setting.
> - We further highlight that all existing SOTA methods, including strong commercial systems trained on extensive internal data (e.g., **Kling**), fail to produce high-quality long try-on videos without inconsistency or degradation. Our method is, to our knowledge, the first to satisfy all key requirements in this space.
> - We would welcome clarification on which “existing techniques” the reviewer believes are merely being combined, so we may better address the concern.
>
> > I don't think it is appropriate to claim "implicit 3D-aware" generation, which also appears confusing in Figure 4. Any well-trained video generative model inherently exhibits such properties to some degree.
>
> - We appreciate the opportunity to clarify this point. The term “**implicit 3D-aware generation**” in our paper (Fig. 4(b), the “cloud” portion) does *not* refer to the general notion of “3D-consistent video generation” or “3D world modeling” that some video generative models may exhibit to a limited extent.
> - Specifically, **in our context**,  “implicit 3D-aware generation” refers to the following:
>   - Our **anchor video**, as shown in Fig. 3(b), can be reconstructed into a 3D form and, thus, can be regarded as a **pseudo-3D representation** (L162).
>   - Our generation process is *explicitly conditioned* on this pseudo-3D anchor video, enabling our model to preserve consistent user and garment appearance across time—what we refer to as *implicit 3D-awareness* (L164).
>   - As other video generative models do *not* condition on such anchor videos, they do not support the “implicit 3D-aware generation” described in our work.
> - More broadly, even leading commercial models do not reliably achieve 3D or temporal consistency, as illustrated by **Kling’s** performance in Fig. 6. In contrast, our method is the only one, to our knowledge, that enforces such consistency through **anchor-based conditioning**.

---

> ### Comment · Reviewer_6KVB · 2025-08-04
>
> Thanks to the authors for their response. My concerns about novelty have been addressed. The proposed techniques, e.g. anchor video mechanism, demonstrate clear improvements in long video virtual try-on quality by enhancing global temporal consistency, where general approaches like FramePack fall short.
>
> However, I remain unconvinced about the terminology "implicit 3D-aware generation." As it creates confusion with established implicit 3D representations (e.g. Signed Distance Functions (SDFs), Occupancy Functions, and Neural Radiance Fields (NeRFs)) in computer graphics and vision. Using video conditioning as a proxy for 3D information does not constitute true implicit 3D representation. I would suggest reconsidering this terminology to avoid misleading readers who might expect actual implicit 3D modeling, when the method primarily relies on video-based conditioning for spatial awareness.

---

> > ### Author Response · Authors · 2025-08-04
> >
> > We sincerely appreciate the follow-up from reviewer 6KVB. We are glad that the concerns about novelty have been addressed.
> >
> > We appreciate the reviewer’s thoughtful suggestion regarding the terminology “implicit 3D-aware generation.” As the reviewer pointed out, our approach leverages video conditioning as a proxy for 3D information, which motivated our initial use of the term “3D-aware.” This is distinct from the established line of work on *implicit 3D representations* such as SDFs or NeRFs. In fact, even state-of-the-art video generative models do not exhibit such implicit 3D structures.
> >
> > To prevent potential confusion, we will revise the terminology to *“pseudo-3D-aware generation”* or *“proxy-3D-aware generation”* and explicitly clarify its distinction from implicit 3D representations in the revised manuscript.

---

> > > ### Comment · Reviewer_6KVB · 2025-08-05
> > >
> > > Thanks. My concerns are addressed. I will raise my score accordingly.

---

> > > > ### Author Response · Authors · 2025-08-06
> > > >
> > > > We sincerely thank the reviewer for the positive feedback and for raising the score – we are delighted that all the concerns have been addressed. Your constructive comments and suggestions have been invaluable in improving the paper.

---

### Official Review · Reviewer_BXqh · 2025-06-29

**Clarity:** 2
**Significance:** 2
**Originality:** 2
**Rating:** 3
**Confidence:** 4

**Summary:**

This paper presents Virtual Fitting Room (VFR), an auto-regressive video generation framework capable of synthesizing minute-scale virtual try-on videos. By generating videos segment-by-segment, VFR avoids the computational burden of end-to-end long video synthesis. It addresses key challenges—local smoothness and global temporal consistency—via prefix video conditioning and an anchor 360° video capturing full-body appearance. The method demonstrates good performance under diverse motions.

**Questions:**

1. The necessity and efficiency of the repeated refinement steps are not well justified. It would be helpful to report how the number of refinement iterations is determined, the computational overhead introduced, and the potential performance impact if the refiner is applied to other datasets such as Dress&Dance.
2. During the generation process, how much time is spent on the VFR module versus the refiner?

**Ethical Concerns:**

["NO or VERY MINOR ethics concerns only"]

**Final Justification:**

I appreciate the authors’ detailed response. After carefully reviewing all the clarifications, I have decided to maintain my original score. My main concerns remain as follows:

First, the paper does not provide quantitative comparisons against other state-of-the-art methods, and the qualitative evaluation is limited to a single video sequence. This makes it difficult to convincingly demonstrate the advantages and significance of the proposed approach.

Second, several important details remain unclear. For instance, it is not well explained how the anchor video is generated (e.g., if SMPL is not used, what alternative conditioning is applied, such as DWPose?). The authors referred me to the Dress&Dance [8] work for further clarification, but this reference points to an anonymous under-review paper. Moreover, the provided link appears to be broken, and I was unable to access the PDF, which makes it difficult to obtain the necessary information, \eg, the CondNets, the dataset they used.

**Limitations:**

Yes

**Quality:**

2

**Strengths And Weaknesses:**

Strength：
1. An auto-regressive virtual try-on method is proposed, capable of generating temporally consistent videos of arbitrary length.
2. A 3D prior is introduced as an anchor video, enabling the generation of 360-degree videos with 3D consistency.


Weakness:
1. The motion of garments appears overly constrained by the underlying 3D human model in the anchor videos （might be SMPL). For instance, loose clothing exhibits minimal dynamics or moves as a rigid body, failing to realistically simulate cloth-specific behaviors such as the flow of skirts.
2. Several implementation details are unclear. It appears that the anchor video is generated by the proposed VFR model conditioned on a given A-pose. However, Section B.1 of the supplementary material only mentions conditioning on the prefix video and anchor video for DIT. It remains ambiguous how the initial anchor video is produced.
3. The paper does not describe the process of fusing a target user with a specified garment to produce the initial try-on appearance, which is crucial for practical applicability.
4. The inference pipeline lacks clarity regarding the generation of the first frame and the initial prefix video. It is important to clarify whether these are produced by a separate model or integrated into the VFR pipeline, and how much they affect overall video quality.

---

> ### Author Rebuttal · Authors · 2025-07-31
>
> > The motion of garments appears overly constrained by the underlying 3D human model in the anchor videos （might be SMPL). For instance, loose clothing exhibits minimal dynamics or moves as a rigid body, failing to realistically simulate cloth-specific behaviors such as the flow of skirts.
>
> - We clarify that we do *not* use any underlying 3D human model such as SMPL. The anchor video is a generated virtual try-on sequence at a fixed A-pose and under a rotating camera trajectory (L158), rather than being based on any explicit 3D mesh or rigged model.
> - Although the anchor video itself is relatively static, our final generated videos are capable of producing reasonable garment-specific dynamics. Examples include the flow of the black skirt and the checkered skirt (SV3:25), as well as the movement of the yellow trousers (SV4:14).
>
> > Several implementation details are unclear. It appears that the anchor video is generated by the proposed VFR model conditioned on a given A-pose. However, Section B.1 of the supplementary material only mentions conditioning on the prefix video and anchor video for DIT. It remains ambiguous how the initial anchor video is produced.
>
> - During inference, the anchor video is generated using our VFR model, conditioned on the garment image, user image, motion prompt, and A-pose template. Since the anchor video is a short (5s), low-FPS video, it is produced via a **non-autoregressive** diffusion process. This avoids the inconsistency often observed in AR generation and does *not* require conditioning on another prefix or anchor video.
>
> > The paper does not describe the process of fusing a target user with a specified garment to produce the initial try-on appearance, which is crucial for practical applicability.
>
> - We refer the reviewer to our prior work **Dress\&Dance [8]** (Sec. 3) for details on how a target user and a specified garment are fused. This is achieved using *CondNets* (Sec. 3.1) for multi-modal conditioning, and a *garment warm-up training* phase (Sec. 3.2) to enhance garment-user fusion capability. Our method adopts this architecture with minimal modifications.
> - When accessing the corresponding repository for [8], please use the **“Download Repository”** button to obtain the full codebase, instead of downloading individual files, to avoid the bugs of anonymous GitHub.
>
> > The inference pipeline lacks clarity regarding the generation of the first frame and the initial prefix video. It is important to clarify whether these are produced by a separate model or integrated into the VFR pipeline, and how much they affect overall video quality.
>
> - The first frame does not require separate generation. The initial prefix video is generated using the same VFR model without providing a prefix condition (L135). There is no separate model involved—our entire generation pipeline, including the generation of the first segment and the anchor video, uses a single VFR model with variable conditioning inputs.
>
> > The necessity and efficiency of the repeated refinement steps are not well justified. It would be helpful to report how the number of refinement iterations is determined, the computational overhead introduced, and the potential performance impact if the refiner is applied to other datasets such as Dress\&Dance.
> > During the generation process, how much time is spent on the VFR module versus the refiner?
>
> - The necessity of immediate refinement is demonstrated in our ablation study (Suppl. L77, Suppl. Tab. C.1 “Ours NR”, SV4:56), where removing refinement leads to notable degradation over long videos.
> - Our refiner is fine-tuned from a partially trained VFR and takes both the prefix and anchor video as input. Therefore, it would not be fair to apply the same refiner to baselines, which lack these conditioning signals. Without these two conditions, the refinement could mitigate some smoothness issues in the baselines, but the fundamental inconsistency would likely persist.
> - Our generation uses a fixed **2-step refinement** strategy. Since refinement operates on an already generated video, it requires fewer denoising steps than the initial generation. On a single GPU, refinement takes approximately **30–40 minutes**, which is about **one-third the time** required for the main video generation. Altogether, refinement accounts for roughly **one-quarter of the total generation time** (~2 hours; see L290).

---

> > ### Comment · Area_Chair_jvMK · 2025-08-05
> >
> > Dear Reviewer BXqh:
> >
> > Can you share your feedback to authors' responses, either in the discussion thread or the final review?
> >
> > Best, AC

---

> > > ### Comment · Area_Chair_jvMK · 2025-08-07
> > >
> > > A gentle ping on the feedback. --AC

---

> > ### Comment · Reviewer_BXqh · 2025-08-07
> > **Official Comment by Reviewer BXqh**
> >
> > Thank you for the detailed explanations regarding my questions. However, after reading the other reviews and the authors’ responses, I decide to maintain my original score.

---

> > > ### Author Response · Authors · 2025-08-07
> > >
> > > Thank you very much for your thoughtful consideration and for reviewing our detailed responses.
> > >
> > > We would greatly appreciate any *specific* remaining concerns, suggestions, or parts of our response that you may have disagreed with. We would be grateful for the opportunity to clarify or discuss them further during the discussion period.

---

> > > ### Comment · Area_Chair_jvMK · 2025-08-07
> > >
> > > Dear Reviewer BXqh:
> > >
> > > Thanks for sharing your feedback.
> > >
> > > However, it is important for us if you could share why you think you would like to keep your original score, to faciliate the decision process. Particularly, what is the main remaining issues of this work, either from you, or other reviewers' feedback.
> > >
> > > Best,
> > > AC

---

### Official Review · Reviewer_RytG · 2025-07-01

**Clarity:** 3
**Significance:** 2
**Originality:** 2
**Rating:** 2
**Confidence:** 3

**Summary:**

This paper introduces Virtual Fitting Room (VFR) for creating arbitrarily long, high-resolution virtual try-on videos from a single source image of a person and a garment. The method operates auto-regressively, generating the video in sequential segments. To address the key challenges of this approach, the authors propose two core mechanisms: (1) a "prefix video" condition to ensure local smoothness and seamless transitions between adjacent video segments, and (2) an "anchor video," a 360° rendering of the subject in an "A" pose, which serves as a global reference to maintain temporal consistency of the person and garment's appearance throughout the long video. A notable finding is that by enforcing temporal consistency, the model implicitly learns 3D structure, enabling free-viewpoint rendering without any 3D supervision.

**Questions:**

Please see the weaknesses.

I think this paper's scope is not suitable for NeurIPS, and it will also be better to conduct the experiments on public benchmarks instead of private dataset that is not publicly available, given that this paper has no dataset contribution claim.

**Ethical Concerns:**

["NO or VERY MINOR ethics concerns only"]

**Final Justification:**

The reviewer appreciate the effort of author rebuttal. However, the rebuttal does not provide persuasive quantitative results on public benchmarks, so I will maintain my original rating as clear rejection.

**Limitations:**

Yes.

**Quality:**

3

**Strengths And Weaknesses:**

**Strengths**
1. The paper presents a method to generate minute-scale, high-resolution (720×1152) virtual try-on videos from a single image, which the authors claim is the first of its kind. The proposed "anchor video" concept is an intuitive and effective solution to the critical problem of maintaining global temporal consistency in auto-regressively generated long videos.

2. A significant finding is that the model learns 3D-consistent representations as a by-product of enforcing temporal consistency, allowing for 3D reconstruction and free-viewpoint rendering without explicit 3D data.

**Weaknesses**

1. The paper does not contain any quantitative comparisons against baseline methods in the main text. The only quantitative results in the supp. mat. is actually an ablation study. So it is questionable whether this paper is suitable as a submission to NeurIPS? Maybe it will better to submit it to Siggraph or other conferences.

2. The model is trained on a dataset named "Dress&Dance" which is cited as being "Under Review" and linked to an anonymous URL.  This makes it unclear whether the strong qualitative results stem primarily from the proposed VFR method or from the quality of the specific, inaccessible training data. Since the dataset is cited separately, the reviewer will assume that this paper has no dataset contribution claim. In this context, it is less persuasive that the proposed method contributes to the final performance rather than the private data. Besides, the anonymous URL is broken so the reviewer cannot download pdf files containing descriptions of the dataset.

3. Since the anchor video in this method is vital, it is also unclear how the proposed method could ensure or improve 3D consistency in its training. This weaknesses is also related to the second one, where the authors fail to elaborate the details of training set, therefore it is unclear how they promote the video quality and 3D consistency when they train the model to generate the anchor video.

---

> ### Author Rebuttal · Authors · 2025-07-31
>
> > The paper does not contain any quantitative comparisons against baseline methods in the main text. The only quantitative results in the supp. mat. is actually an ablation study. So it is questionable whether this paper is suitable as a submission to NeurIPS? Maybe it will better to submit it to Siggraph or other conferences.
>
> - We clarify that Table C.1 in our supplementary material presents quantitative comparisons against two baselines: **D\&D+FramePack** and **D\&D+Kling**.
> - As the benchmark is large, the evaluation of these baselines is resource-intensive. Thus, we focused on the most representative and competitive baselines (e.g., **Kling**, a commercial product trained on extensive internal data).
> - As there are currently no existing methods for *long video virtual try-on*, these composed baselines and our ablation variants represent the only viable points of comparison.
> - We will move Table C.1 to the main paper and include additional baselines in our revision.
>
> > The model is trained on a dataset named "Dress\&Dance" which is cited as being "Under Review" and linked to an anonymous URL. This makes it unclear whether the strong qualitative results stem primarily from the proposed VFR method or from the quality of the specific, inaccessible training data. Since the dataset is cited separately, the reviewer will assume that this paper has no dataset contribution claim. In this context, it is less persuasive that the proposed method contributes to the final performance rather than the private data. Besides, the anonymous URL is broken so the reviewer cannot download pdf files containing descriptions of the dataset.
>
> - We kindly remind the reviewer that **our most powerful baselines**—D\&D+FramePack and D\&D+Kling—use the **same** Dress\&Dance (D\&D) dataset as our VFR model. This demonstrates that the private data *alone* is not sufficient to produce high-quality long try-on videos.
> - Public video virtual try-on datasets, such as **VVT** (256×192) and **ViViD** (832×624, highly compressed), lack the resolution and fidelity needed for generating minute-scale outputs with detailed garment appearance. As such, the D\&D dataset (≥1080p) is the only viable option for our high-resolution task.
> - The inaccessibility of the anonymous GitHub link is caused by a known bug in the `tdurieux/anonymous_github` repository (see GitHub issues \#395, \#404), and is unrelated to our submission.
>   - As a reliable workaround, we kindly ask the reviewer to use the **“Download Repository”** button on D\&D’s anonymous GitHub page to bypass the bug and access the full codebase, paper, and videos.
>
> > Since the anchor video in this method is vital, it is also unclear how the proposed method could ensure or improve 3D consistency in its training. This weakness is also related to the second one, where the authors fail to elaborate the details of the training set, therefore it is unclear how they promote the video quality and 3D consistency when they train the model to generate the anchor video.
>
> - Since the anchor video is a short, low-FPS sequence, its generation does *not* require conditioning on another anchor video or a prefix. Instead, during inference, the anchor video is produced using our VFR via a single-step, non-autoregressive diffusion process. This avoids temporal inconsistency (which only happens in autoregressive generation) and does not rely on prefix conditioning.
> - To support anchor video generation during training, we randomly drop both the anchor video and prefix conditions with a certain probability. This simulates the setting of anchor video generation. The dropout is handled by the **CondNet** conditioning mechanism (D\&D Sec. 3.1), which passes an empty sequence of tokens to the transformer blocks when conditions are absent.
>   - Please refer to the D\&D paper, included in the repository accessible via the **“Download Repository”** button.
>
>
> > I think this paper's scope is not suitable for NeurIPS, and it will also be better to conduct the experiments on public benchmarks instead of private dataset that is not publicly available, given that this paper has no dataset contribution claim.
>
> - We emphasize that our performance is not attributable to the dataset itself, as both **D\&D+FramePack** and **D\&D+Kling**—trained on the same data—fall short in video quality and consistency. Our ablation studies further highlight the contributions of our novel designs.
>
> > The authors did not address the limitations and potential negative societal impact of their work.
>
> - We have outlined the limitations and potential societal impacts of our method in **L288–L299**, which have also been *acknowledged by other reviewers*.

---

> ### Comment · Area_Chair_jvMK · 2025-08-05
>
> Also, FYI, the evaluation of the effectiveness of a method can be diversed. There is no written rule that a NeurIPS paper has to have a metric evaluation.
>
> Therefore, no numerical evaluation itself cannot be justifiable argument for not "suitable as a submission to NeurIPS".
>
> --AC

---

> ### Comment · Reviewer_RytG · 2025-08-05
> **Response to Author Rebuttal and AC Message**
>
> Thank you to the authors for their detailed rebuttal and to the Area Chair for the follow-up and clarification. I have carefully reconsidered my review in light of the discussion.
>
> My primary concern remains the paper's lack of rigorous and verifiable evaluation. While I acknowledge the AC's point that quantitative metrics are not a strict requirement for a NeurIPS paper, their absence becomes a critical flaw when combined with the use of a private, inaccessible dataset. For example, in 3D asset generation, it is also hard to quantitatively compare results from different methods, but most of them could clearly state their pretrained model (e.g., Stable Diffusion 1.5) and/or the dataset (e.g., Objaverse). It is easier for researchers to compare specific cases qualitatively, where no quantitative metrics in a submission is acceptable.
>
> The authors' argument that their baselines also used the "Dress&Dance" dataset does not fully resolve this. It demonstrates the method's superiority within a closed ecosystem of baselines (D&D+FramePack and D&D+Kling) but prevents the broader research community from independently verifying the results or comparing them against other present or future methods. This fundamentally limits the paper's contribution and impact. The reliance on a private dataset for a newly-defined task, without providing any results on established public benchmarks (even if acknowledged as lower-resolution), makes the claims difficult to situate within the existing literature. For this submission, the result is convincing only in two situations: 1) after the private dataset is officially public as a benchmark for the community, compare this submission with other baseline methods in this benchmark or 2) the authors could show performance comparison with other baseline methods in existing low-resolution benchmarks, e.g., VVT and ViViD.
>
> For these reasons, the work lacks the standard of reproducibility and verifiable evidence expected for a publication at NeurIPS. Therefore, I will maintain my original rating of reject.

---

> > ### Author Response · Authors · 2025-08-06
> > **Response to Reviewer RytG (1/2)**
> >
> > We sincerely appreciate the thoughtful follow-up from Reviewer RytG and the continued engagement during the discussion phase.
> > First, we are glad that a key concern has been addressed – specifically, that our performance gains are attributed to our proposed method rather than the training dataset "Dress&Dance (D&D)." This reinforces the core contribution of our paper, which lies in the method itself. As the reviewer correctly noted, our work does *not* claim any dataset contribution. The reviewer’s remaining concerns focus on whether our results are reproducible and verifiable by others, and on how our work benefits the broader research community. We address each of these points in detail below.
> >
> > -----
> >
> > ### Reproducibility, Verifiable Evaluation, and “Ecosystem”
> >
> > **We will publicly release our codebase, trained model, and the proposed evaluation benchmark** – the *four-level evaluation protocol* (Sec. 1, L42–L53). This will ensure that our results are fully verifiable and reproducible, and will allow other researchers to evaluate their methods using our protocol. Additionally, this release will enable them to readily apply our method to future datasets or their own data. We believe this openness enables broad adoption and impact, ensuring that our contributions benefit the wider research community rather than being confined to a closed ecosystem.
> >
> > ------
> >
> > ### Evaluation Rigor
> >
> > We believe that our evaluation is rigorous, comprehensive, and consistent with the established practice in the field.
> >
> > - We prioritized qualitative results and further introduced a structured four-level visual evaluation protocol as part of our contributions. This emphasis on qualitative evaluation is *consistent with* prior work in video try-on and aligns with practices in related fields such as 3D asset generation, as also noted by the reviewer.
> >
> > - As clarified in the rebuttal, we also provided quantitative comparisons in Suppl. Table C.1 against two strongest baselines trained on the same dataset (D&D+FramePack, D&D+Kling). To better address the reviewer’s concern, we will move this table into the main paper and expand the comparative analysis in the revision.
> >
> > - We would like to clarify that the quantitative comparisons were initially placed in the supplementary material because, in the context of long video virtual try-on, commonly used quantitative metrics (e.g., VFID, FID, CLIP), as adopted in prior work (e.g., ViViD, FashionVDM), fail to capture critical factors such as user and garment appearance fidelity – since they *are agnostic to* the input user or garment images. In fact, designing reliable and meaningful quantitative metrics for video virtual try-on remains a fundamental open question, which is beyond the scope of this work.
> >
> > - We conducted fair comparisons against strong baselines **trained on the same dataset** – both D&D+FramePack and D&D+Kling perform worse than ours in terms of garment fidelity and motion coherence. This demonstrates that our performance gains stem from *methodological innovations* rather than the dataset itself. We believe this validates the contributions of our method independently of the dataset.
> >
> > ------
> >
> > ### On the Use of Private Datasets
> >
> > We respectfully disagree that the use of a private dataset makes this work unsuitable for NeurIPS. We would like to emphasize the following:
> >
> > 1. We would like to clarify that the Dress&Dance (D&D) dataset is not a contribution of this paper, and we make no claims regarding its release. **Decisions about the release and format of D&D fall under the scope of a separate under-review submission**. As such, we believe it would be inappropriate to evaluate or penalize this work based on assumptions or expectations related to *another paper. **Our contributions – namely, the proposed task and methodology – stand independently of the dataset’s availability and remain valid regardless of its public release**.
> >
> > -------
> >
> > [Continued in 2/2]

---

> > ### Author Response · Authors · 2025-08-06
> > **Response to Reviewer RytG (2/2)**
> >
> > [Continued from 1/2]
> >
> > -------
> >
> > 2. **Precedent exists for highly impactful NeurIPS-accepted papers using unreleased private datasets**, even for novel tasks. Examples include:
> >
> >    - Saharia et al., *“Photorealistic text-to-image diffusion models with deep language understanding”* (Imagen), NeurIPS 2022. OpenReview: https://openreview.net/forum?id=08Yk-n5l2Al
> >       - > “We train on a combination of *internal datasets*, with ≈ 460M image-text pairs.” (Sec. 4.1)
> >       - The authors **explicitly declined** to release code or data, as documented in the Ethics Review section.
> >       - In the author–reviewer discussion, they clearly stated:
> >       > “We refer the reviewer to our Limitations and Societal Impact section regarding our considerations for **not releasing the code**.”
> >       - Nonetheless, none of the reviewers nor the AC objected, and the paper was accepted.
> >
> >     - Ho et al., *“Video Diffusion Models”*, NeurIPS 2022. OpenReview: https://openreview.net/forum?id=f3zNgKga_ep
> >       - > “We employ a dataset of 10 million captioned videos.” (Sec. 4.3)
> >       - In their discussion with Reviewer EcMx, the authors even **emphasized the superiority of their private dataset**:
> >         > “The *private dataset* we consider … is **even more diverse**, also including many scenes without people in them. Unfortunately the latter **cannot be made public**.”
> >       - Again, no reviewer or AC raised any concerns, and the paper was accepted.
> >
> > These examples demonstrate that **NeurIPS has a clear history of accepting top-tier work built on private datasets**, provided that:
> > - The scientific contribution is clear and impactful,
> > - The results are verifiable through released model/code or sufficient experiments, and
> > - The community can benefit from the ideas, even if the training data itself is inaccessible.
> >
> > We believe our submission satisfies all of the above:
> > - Our code, model, and evaluation benchmark *will be released*;
> > - We conduct *carefully controlled comparisons* on the same dataset;
> > - And we propose *a novel problem formulation and solution*, likely to inspire follow-up work.
> >
> > In light of these, we respectfully request that the use of a private dataset alone should not be used as a reason for rejection – just as it was not for the papers above.
> >
> > ------
> >
> > > “For this submission, the result is convincing only in two situations: 1) after the private dataset is officially public as a benchmark for the community, compare this submission with other baseline methods in this benchmark”
> >
> > We respectfully disagree with the reviewer’s assessment. As clarified earlier, decisions regarding the release and format of the Dress&Dance (D&D) dataset fall under the scope of a separate under-review submission. Our contributions – specifically, the proposed task and methodology – are clearly defined and demonstrate strong performance gains over competitive baselines, independent of the dataset’s availability.
> >
> > We believe it would be counterproductive to delay the dissemination of this work, especially given the relevance and timeliness of long-video virtual try-on. Holding back this contribution risks slowing down progress in the field, including the development of future methodologies and datasets that could build upon our framework.
> >
> >
> > > “For this submission, the result is convincing only in two situations: 2) the authors could show performance comparison with other baseline methods in existing low-resolution benchmarks, e.g., VVT and ViViD.”
> >
> > We respectfully disagree with the reviewer’s assessment. As clarified in the rebuttal, while public datasets such as VVT and ViViD are available, they suffer from **low resolution** and **short duration**, making them **unsuitable for evaluating long, high-fidelity virtual try-on**, which is the core focus of our work.
> >
> > Therefore, we believe that forcing evaluation on these datasets would not meaningfully serve the community, as their limitations prevent them from capturing the real-world challenges and goals central to our task.
> >
> > ------
> >
> > **In summary**, we sincerely ask the reviewer to assess the paper holistically: Our **technical contributions are clear**, our **evaluation protocol will be made public**, and our **code and model will be released** – all of which ensure the reproducibility, verifiability, and broader impact of our work. The decision to release the training dataset lies outside the scope of this submission. Importantly, our methodological rigor and contributions stand **independent of this dataset**, and we believe this should not be grounds for rejection – particularly given that multiple impactful NeurIPS papers have followed a similar structure.
> >
> >
> > We remain open to further clarifying any point. Thank you again for the thoughtful discussion and careful consideration.

---

> > > ### Comment · Reviewer_RytG · 2025-08-06
> > >
> > > I thank the authors for their further clarification and engagement in this discussion. While I appreciate the reference to prior work such as Imagen, I believe this comparison highlights a key distinction. Although Imagen was trained on a private dataset, its performance was substantiated with robust quantitative results on public evaluation benchmarks (Tables 1 & 2). This practice is widely accepted because it provides verifiable evidence, either qualitatively or quantitatively. The current submission, however, lacks a comparable evaluation on public evaluation benchmarks. Therefore, I must respectfully disagree with the authors' claims regarding the reproducibility and verifiability of their work. I hope the Area Chair will take this justification into consideration.

---

> > > > ### Author Response · Authors · 2025-08-06
> > > > **Our Proposed Evaluation Benchmark Will Be Released**
> > > >
> > > > We thank the reviewer again for the thoughtful response. We are glad that our reply has partially addressed the concerns – particularly regarding the use of a not-yet-public dataset for training.
> > > >
> > > > The remaining concern appears to be the lack of evaluation on public benchmarks. Regarding this, we would like to further clarify:
> > > >
> > > > - As also acknowledged by the reviewer, our paper introduces a “newly-defined” task: long video virtual try-on. *Consequently*, there is **no existing public benchmark appropriate** for evaluating this task. This is a **fundamental difference in task setting** compared to works like *Imagen* (Saharia et al.), which operate within well-established tasks with existing public benchmarks. We believe this distinction is critical and should be considered when evaluating the reproducibility and impact of our work.
> > > >
> > > > - To address this gap, we have **introduced a new evaluation benchmark** that includes a set of long video try-on tasks and a systematic four-level evaluation protocol. We conducted thorough comparative experiments using this benchmark in the paper.
> > > >
> > > > - As clearly stated earlier, we will publicly **release all assets – code, trained models, and the proposed benchmark** – to support full reproducibility and facilitate future research.
> > > >
> > > > - Finally, we note that the creation of new benchmarks for emerging tasks is a **well-established and accepted practice** in the research community and, in itself, should not constitute grounds for rejection.
> > > >
> > > > We elaborate on these points in more detail below to provide additional context.
> > > >
> > > > ---
> > > > ### 1. There is no existing public benchmark appropriate for evaluating the novel task of long video virtual try-on
> > > >
> > > > As stated in our previous response, existing public datasets such as VVT (256×192, short clips) and ViViD (832×624, heavily compressed short clips) were designed for *low-resolution and short-duration try-on*, and thus **cannot capture the core challenges** of our task, which include:
> > > >    - High-resolution garment and identity preservation,
> > > >    - Smooth motion generation,
> > > >    - Long-term temporal consistency.
> > > >
> > > > Our task is *fundamentally different* from short-clip try-on or image-based synthesis, necessitating the development of a new evaluation protocol. As such, we *constructed a four-level benchmark* specifically designed to evaluate these new capabilities. This benchmark, along with our code and models, *will be publicly released*, as clearly stated in our earlier response.
> > > >
> > > > If the reviewer believes there are existing benchmarks that can reasonably serve the evaluation goals of long video virtual try-on – specifically for high-resolution and long duration, we would sincerely welcome concrete suggestions and would be happy to address them in the revision.
> > > >
> > > > ---
> > > > ### 2. We will release all assets to support reproducibility and community extension.
> > > >    We will release the following assets:
> > > >    - The complete codebase,
> > > >    - Model checkpoints,
> > > >    - Our four-level evaluation benchmark.
> > > >
> > > > This ensures that our method is **fully verifiable and reproducible**, and enables other researchers to run, test, and compare our method both on our proposed benchmark and on any future datasets the community may introduce.
> > > >
> > > > ---
> > > > ### 3. Creating new benchmarks for novel tasks is a well-established and accepted practice.
> > > >
> > > > It is common, and often necessary, for authors to introduce new benchmarks when proposing novel tasks in emerging domains. For example:
> > > >
> > > > - Haque, Ayaan et al. *“Instruct-NeRF2NeRF: Editing 3D Scenes with Instructions.”* ICCV 2023, *Oral*.
> > > >     - The authors introduced a *new benchmark* tailored to their proposed task of 3D scene editing, which was released upon acceptance. This benchmark, along with their code, has since been widely adopted in follow-up works such as “Collaborative Score Distillation” (NeurIPS’23), “Posterior Distillation Sampling” (CVPR’24), “ReplaceAnything3D” (NeurIPS’24), etc.
> > > >     - This example clearly demonstrates that the introduction of a new benchmark **does not hinder** reproducibility or verifiability – on the contrary, it often accelerates progress in new research directions.
> > > >
> > > > Our submission follows this well-accepted pattern: we introduce a new task – long video virtual try-on, and contribute the *first comprehensive benchmark* for evaluating it. As stated earlier, we will release all necessary resources, including code, model checkpoints, and the benchmark, to support reproducibility and future research in this area.
> > > >
> > > > ---
> > > > We respectfully maintain that **our work is reproducible, verifiable, and extensible**, and we kindly ask the reviewer to consider it in the broader context of *advancing a new research direction* at the intersection of virtual try-on and long video generation.
> > > >
> > > > If there are additional public benchmarks or evaluation protocols that we may have overlooked, we would sincerely welcome the reviewer’s suggestions and are open to incorporating such perspectives in future updates or extensions of this work.

---

### Official Review · Reviewer_VQN7 · 2025-07-03

**Clarity:** 2
**Significance:** 2
**Originality:** 2
**Rating:** 4
**Confidence:** 4

**Summary:**

This paper introduces the Virtual Fitting Room (VFR), a novel framework for generating arbitrarily long virtual try-on videos from a single user image and a garment reference, which uses an auto-regressive generation process guided by an anchor video to ensure global temporal consistency and local smoothness. Experiments show the efficiency of the proposed method.

**Questions:**

Please refer to weaknesses.

**Ethical Concerns:**

["NO or VERY MINOR ethics concerns only"]

**Final Justification:**

Thanks for authors' detailed response. My concerns are addressed. This work does contribute to the specific field of virtual try-on, but it does not contribute much to general video generation. I will raise my score.

**Limitations:**

Most limitations have been declared by the authors.

**Paper Formatting Concerns:**

NO.

**Quality:**

2

**Strengths And Weaknesses:**

Strengths:
1. This paper claim that they are the first method that can generate arbitrarily long, high-resolution human videos of virtual try-on from a single image. Thus this paper is somewhat innovative in the field of try-on.
2. The proposed evaluation protocol covers multiple aspects of virtual try-on quality, including garment consistency, human-garment interaction, and robustness to arbitrary poses. This provides a standardized way to assess future work in the field.
3. The visual results in the paper and supplementary demonstrate the effectiveness of the proposed method

Weakness:
1. Although this paper has made some progress in the try-on field, the overall technical contribution is still limited. This AR video generation way has also been used in other tasks using video generation model, like StarGen [1]. Regarding the video AR generation, the article can also discuss and compare the AR video generation solution in LCT [2], which can use some KV cache to speed up.
2. I am quite confused about how to get the anchor video during inference, which is a 360 video in the paper. But during training, two clips are sampled from one video. Will there be a gap between the anchor video during training and testing if they are too different
3. The refinement process is not elegant and needs to be performed multiple times, which consumes computing power. Is there a more elegant and end-to-end approach?

[1] StarGen: A Spatiotemporal Autoregression Framework with Video Diffusion Model for Scalable and Controllable Scene Generation
[2] Long Context Tuning for Video Generation

---

> ### Author Rebuttal · Authors · 2025-07-31
>
> > Although this paper has made some progress in the try-on field, the overall technical contribution is still limited. This AR video generation way has also been used in other tasks using video generation model, like StarGen [1]. Regarding the video AR generation, the article can also discuss and compare the AR video generation solution in LCT [2], which can use some KV-cache to speed up.
>
> - First, our *overall technical contribution* is not merely an “AR video generation way.” VFR takes a single image of the user and the garment to produce a **high-resolution, smooth, and temporally consistent long try-on video** where the user performs the desired **motion**. We devise *novel training schemes* that allow our model to be trained using **only short video data**.
>   - This is fundamentally different from StarGen, which focuses on generating videos of a **static** scene.
> - LCT extends full attention mechanisms from individual shots to the scene level by incorporating interleaved 3D positional embeddings and an asynchronous noise strategy.
>   - In contrast, our VFR adopts a *standard and streamlined* conditional diffusion framework with multiple conditioning signals. While VFR currently does not use KV-caching, it could potentially benefit from a similar acceleration strategy (e.g., enabling KV-cache for prefix videos), which we consider a promising direction for future work.
>   - We also note that LCT is a concurrent work and has not been published in any peer-reviewed venue.
>
> - Lastly, neither StarGen nor LCT addresses key challenges such as *over-time performance degradation*, *temporal inconsistency*, and *local smoothness*, as can be observed from the videos available on their respective websites. In contrast, our VFR is specifically designed to mitigate these issues, highlighting its **distinct and effective technical contributions**.
> - We will clarify these points further in the final revision.
>
> > I am quite confused about how to get the anchor video during inference, which is a 360 video in the paper. But during training, two clips are sampled from one video. Will there be a gap between the anchor video during training and testing if they are too different.
>
> - During inference, the anchor video is generated as a **short (5s), low-FPS** video try-on task using our own method (L158), performed via a **single-step**, non-AR diffusion generation. This avoids the temporal inconsistency that only arises in *autoregressive* generation.
> - In training, the model is explicitly optimized to “produce videos that are consistent with the (arbitrarily) given anchor video” (L197). More specifically, because the two sampled clips in training may differ substantially due to occlusions and motion dynamics, the model naturally learns to handle diverse anchor videos, including the A-pose anchor used at inference time.
>
> > The refinement process is not elegant and needs to be performed multiple times, which consumes computing power. Is there a more elegant and end-to-end approach?
>
> - The refinement process plays a key role in preventing performance degradation, as demonstrated in our ablation study (Suppl. L77, Suppl. Tab.C.1 “Ours NR”, SV4:56).
> - As the refinement process operates on an **already generated video** and does not need to generate from scratch, it is substantially easier than the original generation. This allows us to distill it into an efficient one-step denoising model with comparable performance. We leave this optimization and the end-to-end approach as interesting directions for future work.

---

> > ### Comment · Reviewer_VQN7 · 2025-08-07
> >
> > Thanks for authors' response. However, I still think that the technical contribution of the paper has not reached the bar of NeurIPS. This paper uses anchor videos to maintain global consistency. It simply uses the overlap between different segments to perform AR. I think the structure is simple and only works for the try-on task. It is not suitable for tasks with a larger range of movement, where a single anchor video cannot maintain the consistency of the entire scene. Furthermore, the refinement requires multiple rounds, so I don't think this is a good pipeline. Therefore, I maintain my original score.

---

> > > ### Author Response · Authors · 2025-08-08
> > >
> > > Following our earlier response, we would like to offer additional details to further clarify the reviewer’s remaining concerns.
> > >
> > > ---
> > > > This paper uses anchor videos to maintain global consistency. It simply uses the overlap between different segments to perform AR. I think the structure is simple and only works for the try-on task.
> > >
> > > As respectfully clarified earlier, our work is **scoped specifically to the long video virtual try-on setting**, which is already a well-established topic in the NeurIPS community. At no point do we claim to propose a general-purpose video generation method, so applicability to broader video generation tasks should not be a criterion for rejection.
> > >
> > > As noted in our original rebuttal, long video virtual try-on **cannot be simply treated as a subcase** of generic video generation. It presents **unique challenges** – most notably, maintaining both user identity and garment appearance in a compact yet globally consistent form – which our anchor video mechanism is explicitly designed to address. Current state-of-the-art general video generation methods such as FramePack and Kling fail to meet these requirements, as discussed in our main paper.
> > >
> > > While our specific anchor video design may not directly transfer to arbitrary long video generation, we believe it offers valuable insight: leveraging 3D-aware representations to enforce global consistency in video generation. This insight could inspire future approaches beyond the try-on domain.
> > >
> > > ---
> > > > It is not suitable for tasks with a larger range of movement, where a single anchor video cannot maintain the consistency of the entire scene.
> > >
> > > We acknowledge that our method is not specifically optimized for general scene-level consistency under large motion ranges. However, this is not a critical limitation in our target setting. In virtual try-on, maintaining *user* and *garment* consistency is paramount, and our anchor video design is explicitly tailored to address these aspects. While broader movement scenarios present interesting avenues for future work, they are beyond the primary objectives and practical requirements of virtual try-on tasks.
> > >
> > > ---
> > > > Furthermore, the refinement requires multiple rounds, so I don't think this is a good pipeline.
> > >
> > > We thank the reviewer for raising this concern. We would like to clarify that the refinement process is not fundamentally a drawback of the pipeline, and may have been misunderstood. Notably, standard diffusion models also operate through multi-step iterative refinement, which is a widely accepted practice in generative modeling.
> > >
> > > Conceptually, our refinement process is **simple, principled, and efficient**. As described in L217 of the main paper, it consists of a direct continuation of the generation process:
> > >
> > > ```
> > > segment_raw ← main_generator(conditions);
> > > segment_refined ← refiner(conditions, segment_raw).
> > > ```
> > >
> > > This design can be viewed as a straightforward conditional extension and is computationally light.
> > >
> > > Despite its simplicity, the refinement step provides a **practical and effective solution** to a critical problem in long-range generation: over-time degradation. This issue is well-documented in autoregressive long video generation and affects even recent state-of-the-art methods such as LCT and StarGen. Our refinement mitigates such degradation and improves visual quality across the entire sequence.
> > >
> > > Importantly, although multiple refinement rounds are **supported**, in practice we only perform **two steps**, as noted in our original rebuttal. This strikes a strong balance between **efficacy and computational efficiency**, and does not introduce significant overhead.
> > >
> > > Finally, we emphasize that refinement is not essential to preserving the core strengths of our method. Even without refinement, our model maintains strong user and garment appearance consistency, which is the primary challenge in virtual try-on. As shown in the ablation study (SV4:56), the “No Refine” version preserves identity and garment fidelity, with only background artifacts. In contrast, prior methods (SV6:49) fail to maintain consistency in both appearance and identity. The refinement step enhances overall video quality but does not underlie the core success of our pipeline.
> > >
> > > ---
> > > We would be happy to further clarify any points, and we sincerely thank you again for your time and thoughtful consideration.

---

> > > > ### Comment · Reviewer_VQN7 · 2025-08-09
> > > >
> > > > Thanks for authors' detailed response. My concerns are addressed. This work does contribute to the specific field of virtual try-on, but it does not contribute much to general video generation. I will raise my score.

---

> > > > > ### Author Response · Authors · 2025-08-09
> > > > >
> > > > > We sincerely appreciate the reviewer for the positive feedback and for raising the score. We are pleased that all concerns have been resolved, and greatly appreciate the constructive comments and suggestions that helped strengthen our paper.

---

> ### Comment · Area_Chair_jvMK · 2025-08-05
>
> Dear Reviewer VQN7:
>
> Can you share your feedback to authors' responses?
>
> Best,
> AC

---

> > ### Comment · Area_Chair_jvMK · 2025-08-07
> >
> > A gentle ping on this. --AC

---

> ### Author Response · Authors · 2025-08-07
>
> Thank you for your feedback. We respectfully disagree with your assessment for two reasons:
>
> 1. Our work is specifically focused on virtual try-on, a widely accepted topic at NeurIPS, as evidenced by recent papers like PM-Jewelry (NeurIPS'24 Workshop), AnyFit (NeurIPS'24), Greatness in Simplicity (NeurIPS'23), etc. Our VFR is the *first* work to demonstrate a high-resolution, smooth, and temporally consistent long try-on video where the user performs the desired motion. We compare our approach to state-of-the-art methods, including commercial products like Kling 2.0 (trained on massive internal data) and concurrent academic works like FramePack, to highlight its significant advantages.
>
> 2. While the suggested references, such as StarGen and LCT, are recent and important works, their capabilities are more limited compared to ours, as we've detailed in our rebuttal.
>
> We believe that discovering a simple solution and demonstrating its effectiveness on a novel and practical use case is a substantial contribution to the NeurIPS community.

---

### Decision · Program_Chairs · 2025-09-17

**Decision:**

Accept (poster)

**Comment:**

This paper was reviewed by 4 experts in the field. The initial review is mostly leaning toward rejection. Still, after the authors clarified many concerns by the reviewers, two of the reviewers turned positive. The final rating to this work is mixed (rating: 2, 3, 4, 4).

Overall, this is a high-quality work solves an important problem. ACs think that the virtual try-on, problem that authors are trying to solve, is a very important to the community. Moreover, the list of challenges the authors was solving (garment consistency, human-garment interaction, and robustness to arbitrary poses, long-term generation) are important and long standing issues of this field.

Moreover, the visual quality is quite decent, as acknowledge by several reviewers.

Still, there are some issues being raised during the discussion. One concern is the technical contribution, as the proposed solution may be similar to other AR based video generation. During discussion, reviewers agreed that the proposed solution still has some novel design for Virtual Try-On tasks. Although this may limit its usage to other tasks, ACs feel that this still should be considered a novel framework.

Another concern is the quantitative evaluation, as the main paper does not have any quantitative evaluation. The only one is in supplementary material. Although this may limit the fidelity, for video generation, directly inspecting the visual result is still a valid way to evaluate in the community. Therefore, ACs did not think this issue itself is enough to reject this work.

The last one is reproducibility, as the authors are trained on private data. Although NeurIPS encourages release of all code and data, this is not a hard requirement. And evaluation shows in the current version is enough to demonstrate the effectiveness of this work.

Therefore, even though this work has several issues, ACs feel that this is still a high-quality work and would benefit the community.

Considering all these, the decision is to recommend the paper for acceptance to NeurIPS 2025. Also, the authors are expected to release all codebase, trained model, and the proposed evaluation benchmark, as promised in the rebuttal.